# StarTrail: Concentric Ring Sequence Parallelism for Efficient Near-Infinite-Context Transformer Model Training

**Ziming Liu**[*]
National University of Singapore
liuziming@comp.nus.edu.sg

Shaoyu Wang[*]
University of Southern California
wangshao@usc.edu

Shenggan Cheng
National University of Singapore
shenggan@comp.nus.edu.sg

Zhongkai Zhao
National University of Singapore
zhongkai.zhao@u.nus.edu

Kai Wang
National University of Singapore
kai.wang@comp.nus.edu.sg

Xuanlei Zhao
National University of Singapore
xuanlei@comp.nus.edu.sg

James Demmel
University of California, Berkeley
demmel@berkeley.edu

Yang You
National University of Singapore
youy@comp.nus.edu.sg

## Abstract

Training Transformer models on long sequences in a distributed setting poses significant challenges in terms of efficiency and scalability. Current methods are either constrained by the number of attention heads or excessive communication overheads. To address this problem, we propose **StarTrail**, a multi-dimensional concentric distributed training system for long sequences, fostering an efficient communication paradigm and providing additional tuning flexibility for communication arrangements. Specifically, StarTrail introduces an extra parallel dimension and divides the peer-to-peer communication into sub-rings to substantially reduce communication volume and avoid bandwidth bottlenecks. Through comprehensive experiments across diverse hardware environments and on both Natural Language Processing (NLP) and Computer Vision (CV) tasks, we demonstrate that our approach significantly surpasses state-of-the-art methods that support Long sequence lengths, achieving performance improvements of up to 77.12% on GPT-style models and up to 114.33% on DiT (Diffusion Transformer) models without affecting the computation results.

## 1 Introduction

Over the past decade, Transformer[38] models have made remarkable strides in diverse fields, including computer vision (CV) and natural language processing (NLP). As the technology has evolved, the ability to efficiently process long sequences with Transformer has emerged as a pivotal challenge. For instance, in text summarization, the ability to handle extensive sequences is vital, as the content to be summarized can range from lengthy chapters to entire books [17, 3]. Similarly, chat-based applications, such as ChatGPT [1], require the capacity to process extensive dialogue

---

[*]Equal Contribution.

39th Conference on Neural Information Processing Systems (NeurIPS 2025).

histories to ensure conversational consistency. There are also applications in other fields like video generation[5, 30] and protein structure prediction[15, 7].

The long context in the above scenarios has introduced several challenges for model training and inference: 1) **Efficiency and Adaptability**. The challenge of efficiency predominantly lies in handling long sequences that require quadratic computations during attention, and in addressing the large amount of communication during distributed processing. 2) **Memory**. Besides the major obstacle of storing the model weight and optimizer states, the activation has also exceeded the capacity of a single GPU and risen as a new memory challenge due to the extreme sequence length. 3) **Scalability**. Current Transformer models usually require thousands of GPUs for pre-training, even with datasets of regular lengths. For longer sequences, ensuring an acceptable scaling speedup rate with both the sequence length and the number of GPUs increasing is even more critical to reducing time and economic costs.

Traditional parallelisms such as Data Parallelism[12, 37, 22, 41], Tensor Parallelism[37, 39, 40], and Pipeline Parallelism[13, 10, 23, 25] distribute the model, input batch, and the optimizer states, but can not directly address the large memory requirement of extremely long sequences as the sequence length dimension remains unchanged. To break through this obstacle, Sequence Parallelism has been introduced, splitting the input on the sequence length dimension. Mainstream Sequence Parallelism schemes can generally be classified into two categories, and are usually combined [11] to complement each other's drawbacks. Methods like DeepSpeed Ulysses[14], which are based on all-to-all communication, offer efficiency but require the splitting of attention heads. Consequently, these methods are limited in scalability and can not be scaled to more devices than the number of attention heads. On the other hand, peer-to-peer communication methods[24, 20], such as Ring Attention[24], do allow for near-infinite context lengths; however, they require the transmission of complete keys and values across all GPUs, leading to significantly high communication loads. In summary, there remains a deficiency in communication-efficient methods that are capable of supporting near-infinite context lengths. In this paper, we focus on solving the communication inefficiency of ring-style sequence parallelism.

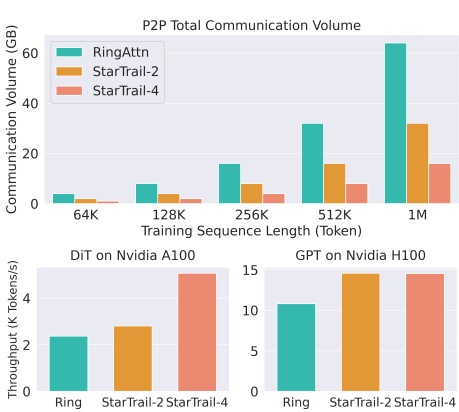

Figure 1: StarTrail-2 and StarTrail-4 theoretically save around 50% and 75% of total P2P communication volume for various sequence length and achieves up to 2x speedup in end-to-end training
.

To solve these challenges, we introduce StarTrail, a novel near-infinite-context Transformer training system with concentric multi-ring sequence parallelism that incorporates an additional parallel dimension into the existing ring-style communication. Specifically, instead of including all GPUs in a single parallel group as done in Ring Attention [24], StarTrail groups the GPUs into teams and divides the peer-to-peer communication within these teams. This approach fosters an efficient communication paradigm and provides extra tuning flexibility for communication arrangements. With very little additional memory cost, StarTrail parallelism significantly reduces the peer-to-peer communication volume, as shown in Figure 1. Compared to previous works, StarTrail is not limited in supported sequence length by attention heads like DeepSpeed Ulysses[14] and Megatron Sequence Parallelism[18], and also shows better communication efficiency and scalability than Ring Attention[24]. We perform experiments on mainstream Transformer models, including GPT-style[33] and DiT-style[31], conducting performance and scaling tests across various computing clusters. Experiment results indicate that our StarTrail system outperforms Ring Attention by up to 77.12% on the GPT model and up to 114.33% on the DiT model, showcasing its efficiency and scalability.

# 2 Background and Related Works

## 2.1 Long Sequence Training and Sequence Parallelism

The key mechanism behind these Transformer-based models is attention[38], which captures the text feature by calculating the attention score between every two single tokens. However, the sequence length can reach hundreds of thousands, when dealing with multi-round chatting, or high-resolution long video generation. It then becomes necessary to distribute the sequence across multiple GPUs. This distribution helps to reduce both the memory and computation demands on any single device. This strategy is also known as sequence parallelism. Presently, Sequence parallelism can be divided into two main categories: attention-head-sharding-based and peer-to-peer-communication-based. The former involves distributing the attention heads of multi-head attention across multiple GPUs, whereas the latter resembles a distributed version of FlashAttention, relying on peer-to-peer communication to transfer keys, values, and intermediate statistics.

### 2.1.1 Ring-peer-to-peer-communication-based

The primary method in peer-to-peer-communication-based strategies is Ring Attention[24], which is also the main baseline of this work. Introduced in 2023, Ring Attention[24] innovatively partitions the sequence dimension and utilizes a ring-style peer-to-peer (P2P) communication pattern to transfer Keys and Values across all GPUs. Each GPU receives the key and value matrices from the preceding rank, updates the local attention score, and then forwards them to the next rank, as is shown in Figure 2. This method employs an online-softmax and updates attention scores incrementally, allowing the computation of attention scores without retaining the full sequence length. Thus, it potentially supports infinite context, provided sufficient

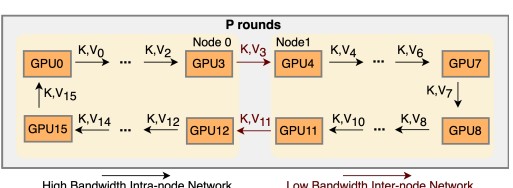

Figure 2: An example of Ring Attention Computation on 16 GPUs in two nodes. The Communication is largely limited by the inter-node bottleneck.

computing resources are available. However, the requirement for the same number of rounds of P2P communication as the number of GPUs renders this approach less efficient in environments with high-latency communication.

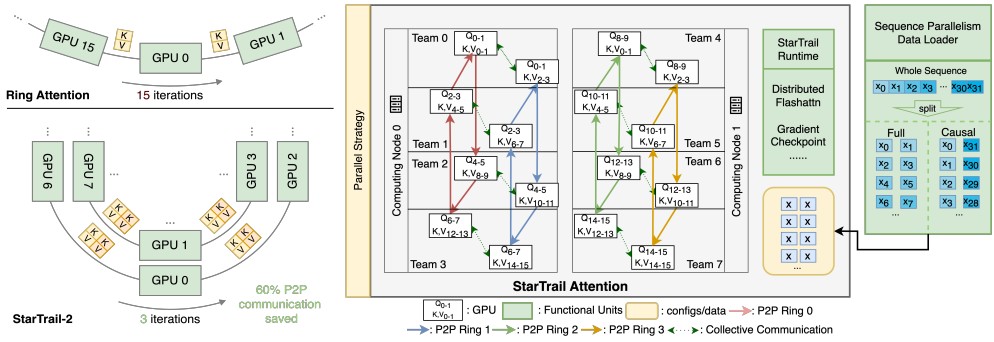

(a) StarTrail-2 reduces 60% P2P communication volume on 16 GPUs compared with ring attention.

(b) StarTrail divides one ring into four concentric sub-rings, with every two connected with collective communication.

Figure 3: An overview of the StarTrail Training System

### 2.1.2 Attention-Head-Sharding-Based

There are two representative methods, DeepSpeed-Ulysses and Megatron Sequence Parallelism in this category. DeepSpeed-Ulysses[14] transitions from sequence parallelism to a method akin to tensor parallelism with two all-to-all communication. It divides the query, key, and value matrices across the attention heads, thereby preserving the original attention computation structure. Megatron Sequence Parallelism[18] focuses on minimizing memory usage and reducing the necessity for

activation recomputation rather than efficiency. The two methods both rely on the number of attention heads to split the sequence, thus limited in scalability, especially when employing techniques like grouped-query attention (GQA) [2] or multi-query attention (MQA) [36]. As these two sequence parallelism methods are **orthogonal** to the ring-based method, they are usually combined with ring attention to enable longer sequences.

In this paper, we focus on optimizing ring-style sequence parallelism, noting that integrating it with DeepSpeed-Ulysses Parallelism does not interfere with the underlying ring process.

## 3   StarTrail Training System

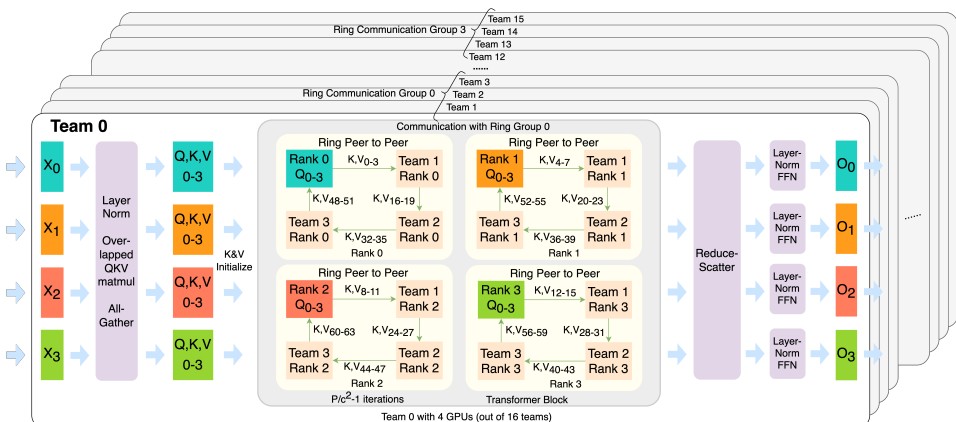

Figure 4: An example of StarTrail Attention on 64 GPUs. Each team member forms a sub-ring with members with the same local rank from other teams in the same ring communication group, reducing the communication volume of each team member by 75%.

### 3.1   Motivation

Through observation, we identify two main drawbacks of Ring Attention. First, the communication overhead is exceedingly high because every GPU in the system must send and receive keys and values for nearly the entire sequence length before completing the attention computation. Second, variations in bandwidth between and within computing nodes can cause communication bottlenecks. As illustrated in Figure 2, the bandwidths between GPUs 3 and 4 and between GPUs 11 and 12 are lower than those between other GPUs in the ring. Despite this, the system is forced to operate in a complete circle, which can result in unnecessary idle times for other GPUs. To solve these drawbacks, we develop the StarTrail training system, which we will detail in the following section.

### 3.2   StarTrail Attention

As discussed in the previous section, a major limitation of Ring Attention is the extensive amount of peer-to-peer (P2P) communication required, which becomes problematic in environments with weak connections between computing nodes. We enhance the ring sequence parallelism by introducing an additional dimension. The fundamental idea of StarTrail is akin to a divide-and-conquer strategy. During attention, each token must compute its attention score with every other token in the sequence. While Ring Attention passes keys and values along a ring of $P$ GPUs over $P-1$ iterations, our approach introduces the concept of a **team**. In this setting, each team member interacts only with a designated portion of the overall sequence, and the results are later aggregated using collective

Figure 5: Meanings of the symbols that are used in this paper

| | |
|---|---|
| $P$ | The number of GPUs |
| $C$ | The parallel size of StarTrail (team size) |
| $H$ | The hidden dimension size of the Transformer blocks |
| $N$ | The total number of tokens within the whole sequence |
| $B$ | The training batch size |
| $W$ | The communication bandwidth between GPUs |
| $L$ | The communication latency between GPUs |

communication. Thus, StarTrail can be devided into three phases: **preprocessing**, **ring-phase**, and **postprocessing**.

For **preprocessing**, we duplicate the queries within a team using an all-gather operation. This ensures that when a GPU receives new keys and values, it can compute the attention scores for the entire team's queries. Similarly, gathering the keys and values allows us to reduce the number of P2P communication iterations by transmitting longer sequences per iteration. Following the preprocessing, we enter the **ring-style communication phase**. With the number of GPUs in one team being $C$, $CN/P$ tokens are exchanged in each iteration, and each GPU is responsible for computing $N/C$ tokens. This leads to a number of iterations of $\frac{N/C}{CN/P} = \frac{P}{C^2}$, within a smaller ring, which we refer to as a **subring**. For convenience, we group $\frac{P}{C^2}$ adjacent teams into a **team group** for subring communication, where GPUs sharing the same local team rank form the subring. An initial P2P communication step is executed to ensure that each team group has access to the complete set of keys and values for the sequence (details are provided in the Appendix). After completing the subring iterations, each GPU holds $1/C$ of the overall computation result for its team. With the help of online softmax, we then apply a simple reduce-scatter to combine these results while eliminating the duplicate tokens, which we refer to as the **postprocessing**. Throughout the attention process, asynchronous communication is employed alongside the early launch of communication kernels to maximize the overlap of computation and communication tasks. Now we will delve into more details in the StarTrail training process.

### 3.2.1 Configurations of StarTrail Parallelism

In the StarTrail system, GPUs are grouped into *Teams* to coordinate computation and communication tasks more efficiently. StarTrail introduces an additional parameter, $C$, which determines the replication factor of the input and, consequently, the number of GPUs within each team. The range of $C$ is from 1 to $\sqrt{P}$. When $C$ equals one, the algorithm falls back to Ring Attention. When $C$ equals $\sqrt{P}$, the algorithm becomes a completely collective-communication-based one with no rings. When $1 < C < \sqrt{P}$, it becomes a structure with multiple rings looping concurrently.

---

**Algorithm 1** StarTrail Attention Block (Forward)

---

**Require:** Input sequence $\mathbf{x}$, Linear Function **query, key,** and **value**, attention parallelism size **c**, global rank **r**, global size **gs**, team process group **pg**, init send/recv target $\mathbf{r}_{\text{send}}$ and $\mathbf{r}_{\text{recv}}$
1: compute the gathered $\mathbf{q}_{\text{team}}, \mathbf{k}_{\text{team}}, \mathbf{v}_{\text{team}} = \text{AllGather\_QKVmatmul}(\textbf{query, key, value, x, pg})$
2: launch the asynchronous send and receive request
   $\textbf{req}_{\text{send}}$ and $\textbf{req}_{\text{recv}}$, sending $\mathbf{k}_{\text{team}}, \mathbf{v}_{\text{team}}$ to $\mathbf{r}_{\text{send}}$ and receiving $\mathbf{k}_{\text{next}}, \mathbf{v}_{\text{next}}$ from $\mathbf{r}_{\text{recv}}$
3: get the ring P2P target $\mathbf{r}_{\text{next}}$ and $\mathbf{r}_{\text{last}}$ with get\_P2P\_ranks(**r, gs, c**)
4: initialize attention score $\mathbf{O}$, extra statistics **lse** to zero. // lse stands for log-sum-exp
5: **for** $1 \leq i \leq$ world\_size$/c^2$ **do**
6:     wait for $\textbf{req}_{\text{send}}$ and $\textbf{req}_{\text{recv}}$
7:     $\mathbf{k}_{\text{current}} = \mathbf{k}_{\text{next}}, \mathbf{v}_{\text{current}} = \mathbf{v}_{\text{next}}$
8:     launch $\textbf{req}_{\text{send}}$ to send $\mathbf{k}_{\text{current}}$ and $\mathbf{v}_{\text{current}}$ to $\mathbf{r}_{\text{next}}$, launch $\textbf{req}_{\text{recv}}$ to receive $\mathbf{k}_{\text{next}}$ and $\mathbf{v}_{\text{next}}$
       from $\mathbf{r}_{\text{last}}$
9:     calculate **lse, O** =
       forward\_iteration(**lse, O**, $\mathbf{q}_{\text{team}}, \mathbf{k}_{\text{current}}, \mathbf{v}_{\text{current}}$)
10: **end for**
11: compute $\mathbf{O}_{\text{final}} = \text{ReduceScatter\_combine}(\textbf{lse, O, pg})$
12: return $\mathbf{O}_{\text{final}}$

---

**Forward Propagation**. In Figure 4, we have an example of one team of four GPUs out of all the 64 GPUs performing StarTrail-style attention. Each training iteration begins with the dataloader splitting the entire input sequence of length $N$ into $N/P$ sub-sequences, which are then loaded onto each GPU. As previously mentioned, the next step involves computing the queries, keys, and values. These are computed separately via matrix multiplication, followed immediately by the launch of the all-gather kernel, which gathers the above QKVs within the team, allowing for the overlap of up to two-thirds of the communication with computation.

Once this phase is complete, each GPU within the team possesses the same Q, K, and V, each of a length of $\frac{CN}{P}$. To distribute the communication and computation tasks among the team members,

we divide the original workload based on four specific ranks assigned to each GPU. These ranks determine each GPU's partners and position within the P2P ring, as is shown in Figure 6.

Following the setup, the Keys and Values are dispatched to their designated locations within the cluster to establish the initial sub-ring, setting the stage for the multi-ring iteration phase of StarTrail attention. Given that each sub-sequence is $\frac{CN}{P}$ long and each GPU is tasked with computing the attention score for $\frac{1}{C}$ of the whole sequence, it results in $\frac{N/C}{CN/P} - 1 = P/C^2 - 1$ rounds of communication. This implies that there are $P/C^2$ GPUs in one ring.

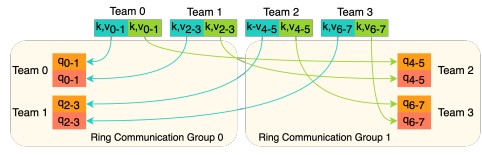

Figure 6: An example of ring initialization process of 8GPUs and 4 sub-rings in Star-Trail.

The iteration process involves storing the log-sum-exp (lse) and intermediate output O, which are updated step by step. Queries are retained locally, while Keys and Values circulate through the ring via P2P communication. After completing the iterations, each team member accumulates the attention scores for the entire team's sub-sequence of Queries with $1/C$ of the Keys and Values from the full sequence.

A simple reduce-scatter operation is then employed to amalgamate the intermediate results and distribute them among the team members. Each GPU ultimately contains the final attention score for its portion of the sequence over the entire sequence.

**Backward Propagation**. The major distinction between backward and forward propagation is the inability to calculate queries independently during the backward phase. Unlike forward propagation, the backward phase requires the complete set of keys and values to calculate the gradient for queries, and vice versa. To manage this, we have structured the gradient calculation into two loops: the key & value outer loop and the query inner loop. In the outer loop, gradients for keys and values are tracked and maintained fixed on the corresponding GPUs within the sub-rings; these gradients do not transfer between GPUs. The inner loop, however, handles the gradients for queries, which start initialized as zero and are circulated along the sub-rings together with the Queries themselves. During each iteration, the approach mirrors the backward computation method used in FlashAttention[8], where the updated gradient of the current query shard is passed to the next GPU in the ring, while the gradients for keys and Values are retained for subsequent query shards.

### 3.2.2 Theoretical Analysis

During the analysis, we will employ a case study using the StarTrail system with an attention parallel size of $C = 4$ on a llama-30B model, which consists of 64 layers. For this model, referred to as model M, the batch size = $B$ is set to 1, the sequence length = $N$ to 65536, the hidden dimension = $H$ to 6656, and the number of GPUs = $P$ to 64. Additionally, the computation will utilize bfloat16 precision.

**Communication Analysis**. Let's analyze the communication overhead within one forward Transformer block on a single GPU. For Ring Attention, the communication is primarily due to the ring P2P loop. As the total number of iterations done is $P - 1$, the total communication overhead can be calculated as:

$$(P-1)(\frac{2BNH}{PW} + L) = \frac{2BNH(P-1)}{WP} + (P-1)L \tag{1}$$

and this overhead can be partially overlapped with the attention computation.

For StarTrail, the communication overhead comes from both collective and P2P. The collective overhead for all-gather and reduce-scatter is:

$$\frac{4BNH(C-1)}{PW} \tag{2}$$

while the P2P communication can be similarly computed as:

$$(\frac{P}{C^2} - 1)(\frac{2CBNH}{PW} + L) = \frac{(P-C^2)2BNH}{CPW} + (\frac{P}{C^2} - 1)L \tag{3}$$

The advantages of StarTrail over Ring Attention during the ring-P2P phase are evident in three main aspects: 1) **Reduced Communication and Latency**: Ring Attention requires C times more communication than StarTrail, significantly increasing the bandwidth requirement across the entire cluster. For the llama 30B model M, the total communication volume of ring P2P communication and collective communication volume for Ring Attention and StarTrail can be computed as 1.625 GB and 0.152 GB(collective) + 0.406GB (P2P) = 0.558GB. Furthermore, while Ring Attention necessitates $P - 1$ iterations per attention block, StarTrail only requires $\frac{P}{C^2} - 1$, reducing the latency overhead by around $C^2$. 2) **Localized Communication**: In scenarios like those depicted in Figure 3, StarTrail's ring P2P communication can be confined within the same computing node, where bandwidth is typically much higher than between computing nodes. Conversely, Ring Attention demands inter-node communication during every iteration, which can be less efficient. 3) **Enhanced Overlap of Communication and Computation**: During each iteration, the communication volume of StarTrail is $C$ times higher than that of Ring Attention, while the computational volume during attention is approximately $C^2$ times greater. This higher computation-to-communication ratio makes it easier for StarTrail to overlap P2P communication with computation, enhancing overall efficiency.

**Memory Analysis**. In this section, we estimate the theoretical peak memory requirements necessary to store the model weights, activations, and optimizer states. Our implementation utilizes the Adam Optimizer [16], bfloat16 precision, and Zero-2 optimization [34]. We name the memory cost for the model and optimizer as $M_{m+o}$. As for the activation, we refer to the size of one single activation of a sub-sequence on one GPU as

$$A = \frac{B \times N \times H}{P} \tag{4}$$

As we use the checkpointing scheme from [20], a model of $Y$ layers needs to save $Y + 1$ activations as checkpoints. Now we calculate the approximate peak memory after Q, K, and V are already calculated and before the attention computation at the last layer of the whole model. For Ring Attention and StarTrail, the peak memories are:

$$PM_{Ring} = M_{m+o} + (Y + 4)A \tag{5}$$

$$PM_{Star} = M_{m+o} + (Y + 3C + 1)A \tag{6}$$

, where C is the StarTrail attention dimension. And for the example model M, the peak memory would be $M_{m+o} + 68A$ and $M_{m+o} + 77A$, and the extra memory cost compared with Ring Attention is a lot less than 13.2%, while the P2P communication volume is reduced by about 75%. In a word, the extra memory cost is acceptable as a tradeoff for the communication reduction.

## 4 Evaluation

Table 1: Cluster and Model Configurations. All GPUs are connected with NVLink with computing nodes.

| GPU | dev. $\times$ node | Mem. (GB) | inter-node bandwidth | Model | #Heads | #Layers | Dim. |
|-----|-----|-----|-----|-----|-----|-----|-----|
| H100 | $8 \times 8$ | 80 | 8*400Gbps InfiniBand | GPT 3B | 12 | 16 | 4096 |
| A100 | $16 \times 2$ | 40 | 100Gbps Ethernet | GPT 7B | 32 | 32 | 4096 |
| A100 | $8 \times 4$ | 40 | 100Gbps Ethernet | DiT 1B | 24 | 24 | 1536 |
| A100 | $4 \times 8$ | 40 | 100Gbps Ethernet | | | | |

The computational resources we use in the experiments include a local Nvidia H100 cluster with eight nodes and three Nvidia A100 clusters, as listed in table 1. We utilize two model types of total three settings, as listed in table 1. For the DiT(Diffusion Transformer) model, we use similar configurations as those in Stable Diffusion 3[9]. We utilize the backbone Diffusion Transformer only, without other components like the text and image encoders. During training, both models use bfloat16 precision and a batch size of 1 to accommodate longer input sequences.

In the evaluation section, we aim to answer three major questions: 1) How much improvement in throughput can StarTrail bring? Additionally, how adaptable is StarTrail to clusters with both good and poor inter-node connections? 2) Is the additional memory cost incurred by StarTrail acceptable considering the throughput improvement it offers? 3) How does StarTrail perform in scenarios of weak and strong scaling? Specifically, does it outperform Ring Attention when scaled to handle longer inputs?

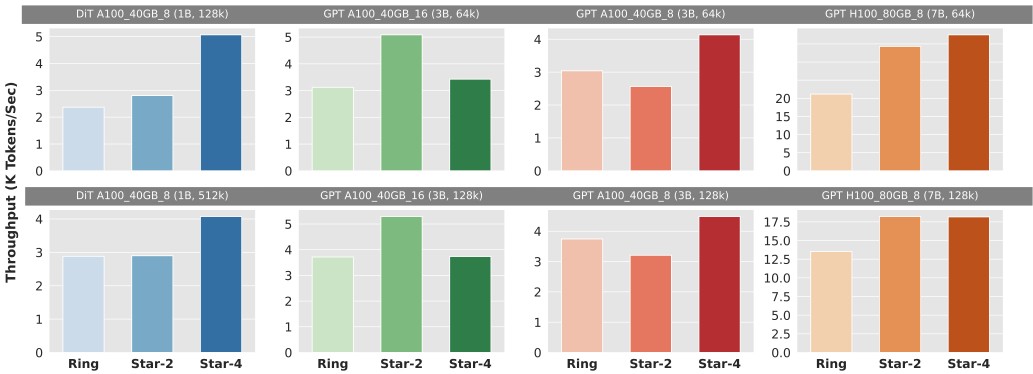

Figure 7: Throughput evaluation of Ring Attention and StarTrail on 32 GPUs from three different clusters. We place the performance of StarTrail with both C=2 and C=4 in the figure. The configurations are marked in the titles of the sub-figures. For instance, A100_40GB_8(1B, 512K) represents that the experiment is on machines with 8 Nvidia A100 40GB GPUs in each node, the model used has one billion parameters, and the sequence length is 512k.

## 4.1 Throughput and Adaptability

Our first experiment aims to assess the performance of StarTrail and Ring Attention across different clusters with varying environments, testing the adaptability of both methods. There are several factors influencing the efficiency of ring-style attention computation:

**Theoretical Computation-Communication Volume Ratio:** Primarily determined by the sequence length used during training. Attention computation exhibits a computational complexity of $O(N^2 \cdot H)$, whereas P2P communication complexity is $O(N \cdot H)$. Thus, the model configuration does not impact this ratio; only the sequence length does. A larger $N$ increases the computation-communication ratio, facilitating easier overlap of communication with computation.

**Compute Capability and Connectivity of GPUs:** The computing overhead, given a specific volume, affects the computation-communication overhead ratio. Higher compute capabilities make overlapping more challenging. We utilize two sets of GPUs in this evaluation: Nvidia A100 40GB and Nvidia H100 80GB, with the latter offering significantly higher theoretical tflops on bf16 computations. Connectivity is considered in two parts: intra-node and inter-node. Our clusters are equipped with NVLink, ensuring robust intra-node communication. For inter-node communication, our H100 nodes leverage InfiniBand with eight adapters per node for superior inter-node bandwidth, whereas the Google Cloud servers use Ethernet. The diversity in node configurations (8-GPU and 16-GPU nodes) allows us to assess adaptability across different topologies. This evaluation not only highlights the inherent differences between the schemes but also tests their flexibility in various hardware settings.

The results of our evaluation are illustrated in Figure 7. We measure throughput in thousands of tokens per second. To better demonstrate how to select the optimal configuration of StarTrail under each condition, we included two configurations, Star-2 and Star-4, in the figure. We omit configurations with lower performance for clarity. As indicated in the figure, in all six settings, at least one configuration of StarTrail achieves higher throughput than Ring Attention, with 2.114x, 1.414x, 1.629x, 1.425x, 1.360x, 1.199x, 1.771x, and 1.346x the throughput of Ring Attention. This advantage is primarily due to the additional parallel dimension that StarTrail introduces. Unlike Ring Attention, which requires inter-node P2P communication in each iteration, StarTrail's P2P communication is mostly confined intra-node, except for initial data transfers. This experiment clearly demonstrates StarTrail's superior performance across various environments. Another observation is that the optimal configuration for StarTrail may vary depending on the environment, reflecting differences in the computation-communication ratio and the trade-offs between collective and P2P communication.

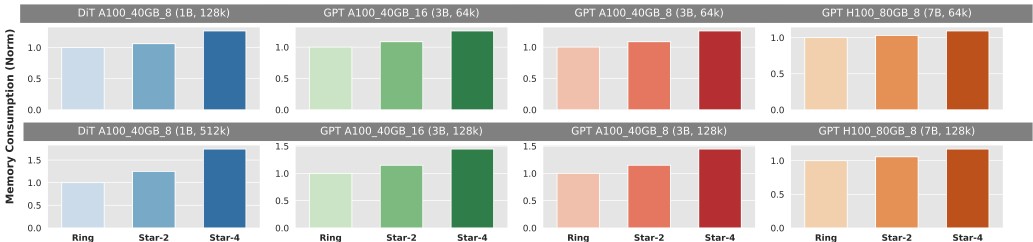

Figure 8: The normalized relative memory cost of different configurations of StarTrail compared with Ring Attention on different clusters.

## 4.2 Memory Consumption

The memory results, displayed in Figure 8, reveal that for the configurations yielding the highest throughput, StarTrail consumes between 7.9% and 30.79% more GPU memory than RingAttention, while achieving 1.199x to 2.114x throughput. Moreover, in scenarios involving larger models, the relative increase in memory consumption due to QKV duplication diminishes. This reduction is explained by equation 6 This phenomenon is further evidenced by the fact that the extra memory ratio for experiments with the 7B model is significantly smaller than that for the 3B and 1B model. Considering the substantial throughput gains provided by StarTrail, this tradeoff between memory usage and efficiency is deemed acceptable.

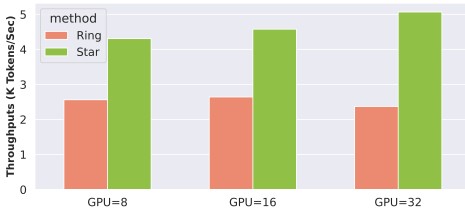

(a) DiT Strong Scaling on Nvidia A100 40GB GPUs. All configurations include inter-node communication.

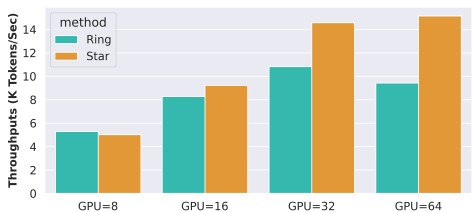

(b) GPT Strong Scaling on Nvidia H100 80GB GPUs.

Figure 9: Strong scaling experiments with fixed sequence length of 128K.

## 4.3 Strong and Weak Scaling

In the scaling tests we carry out experiments for both strong and weak scaling. For strong scaling, we fix the sequence length to 128K while increasing the number of GPUs from 8 to 64 for the GPT model and from 8 to 32 for the DiT model. For weak scaling, we scale the sequence length from 128k to 512k for the DiT model and from 64k to 512k for the GPT model proportionally increasing the number of GPUs from 8 to 32. As is depicted in Figure 9 and 10, StarTrail shows obvious advantage over Ring Attention as we increase the number of GPUs. The results for strong scaling can also explained by the computation-communication ratio. When scaled to more GPUs, the local sequence length on each GPU becomes smaller, and as explained the previous sections, makes it harder to overlap the P2P communication with attention computation. The overall scaling performance is limited by the nature of ring-style communication, but we still consider our improvement over Ring Attention meaningful due to the necessity of using Ring-style Parallelisms during training.

In summary, StarTrail shows better scalability in both strong and weak scaling experiments, making it a better choice for large-scale Transformer model training.

## 5 Conclusion

StarTrail represents an advanced near-infinite-context Transformer model training system, featuring a communication-optimized concentric ring sequence parallelism scheme. Through experiments, we

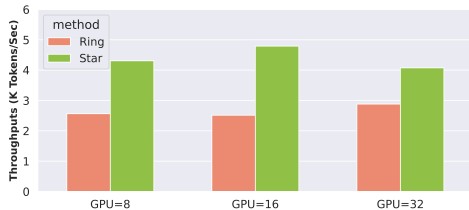 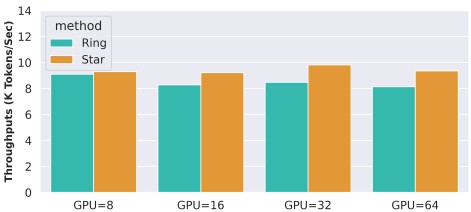

(a) DiT Weak scaling on Nvidia A100 40GB GPUs of sequence length from 128K to 512K. All configurations include inter-node communication.

(b) GPT Weak scaling on Nvidia H100 80GB GPUs of sequence length from 64K to 512K.

Figure 10: Weak scaling Experiments

demonstrate that our system not only achieves high efficiency across various training environments but also excels under both strong and weak scaling conditions for both CV and NLP models. Current limitations of StarTrail include that although orthogonal, we can still further improve the co-design of StarTrail and hybrid parallelism in future works. In an era increasingly demanding longer contexts for both NLP and CV, StarTrail is poised to make significant contributions to the industry and inspire innovative research in academia.

# 6 acknowledgement

Yang You's research group is being sponsored by NUS startup grant (Presidential Young Professorship), Singapore MOE Tier-1 grant, ByteDance grant, NUS ARTIC grant, Apple grant, Alibaba grant, Google Research and Google grant for TPU usage.

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

## A Technical Appendices and Supplementary Material

### A.1 Sequence Parallelism Dataloader

The causal masks present a challenge due to the unbalanced computational load across GPUs: subsequences at the beginning of a sequence require significantly more computation than those at the end. To address this imbalance and achieve load equilibrium among GPUs, we modify the ZigZag scheme introduced by [42], illustrated in Figure 11. The figure illustrates the simplest case of zigzag load-balancing. Notably, the effectiveness of this strategy improves as the number of GPUs increases. This improvement correlates with the expanding difference in computation volume between the first and the last token, which escalates as the sequence length extends. This approach ensures that the total workload on each GPU is balanced, eliminating the need for additional communication mechanisms like those employed in DistFlashAttention[20].

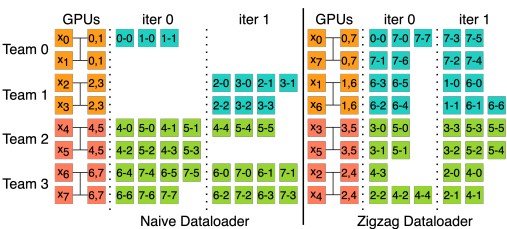

Figure 11: A comparison between naive and zigzag dataloader for 8 GPUs with attention parallel dimension of 2. The corresponding initialization can be found in Figure 6 with the same configuration. The improvement of efficiency from load-balancing increases with the number of GPUs.

## A.2 Details in the training process

Apart from the attention process described in Algorithm 1, we would like do provide a few other details to be more comprehensive. First, before the concentric ring attention, we need to initialize the the Keys and Values and determine each GPU's position within the rings.

Initially, for the setup stage, it is essential to establish the sub-rings by rearranging the activation positions. Specifically, during forward propagation, the queries do not require rearrangement; however, the keys and values must be transmitted to their corresponding positions in the ring prior to commencing the loop. As illustrated in Figure 6 and algorithn, this initialization ensures that each team member holds a different shard of keys and values. Moreover, it guarantees that no two teams within the same ring possess identical keys and values. Initially, for the setup stage, it is essential to establish the sub-rings by rearranging the activation positions. Specifically, during forward propagation, the queries do not require rearrangement; however, the keys and values must be transmitted to their corresponding positions in the ring prior to commencing the loop. As illustrated in Figure 6 and algorithm, this initialization ensures that each team member holds a different shard of keys and values. Moreover, it guarantees that no two teams within the same ring possess identical keys and values.

---

**Algorithm 2** get_init_send()

**Require:** inter-team rank $r_t$, intra-team rank $r_a$, inter-team dimension $d_t$, intra-team dimension $d_a$
  1: team group size = $d_t$ / $d_a$
  2: target team group rank = $r_a$
  3: target team = target team group rank * team group size + $r_t$ // $d_a$
  4: target device intra-team rank = $r_t$ % $d_a$
  5: target global rank = target team * $d_a$ + target device intra-team rank
  6: return target global rank

---

After the initialization of activations, we can set up the rings by providing the GPUs their last and next GPU within their rings, as is described in Algorithm 3.

## A.3 StarTrail Runtime

StarTrail is written in PyTorch[29] and uses the PyTorch torch.autograd.function and NCCL[28] backend for forward and backward implementation. StarTrail also employs multiple techniques during runtime to improve its overall training efficiency.

**Ingetrate Flash Attention.** The StarTrail attention mechanism involves multiple iterations that loop over Keys and Values, with each iteration still using traditional self-attention with corresponding Query, Key, and Value (QKV). This approach enables StarTrail to incorporate flash attention effectively, extending its capability by preserving intermediate states across iterations. Additionally,

**Algorithm 3** get_P2P_config()

**Require:** inter-team rank $\mathbf{r}_t$, intra-team rank $\mathbf{r}_a$, inter-team dimension $\mathbf{d}_t$, intra-team dimension $\mathbf{d}_a$
 1: team group size = $\mathbf{d}_t$ / $\mathbf{d}_a$
 2: self team group rank = $\frac{r_t}{team\ group\ size}$
 3: next team in group = $(r_t + 1)\%$ team group size + team group size $\times$ self team group rank
 4: last team in group = $(r_t - 1)\%$ team group size + team group size $\times$ self team group rank
 5: next device global rank = $r_a$ + next team in group $\times d_a$
 6: last device global rank = $r_a$ + last team in group $\times d_a$
 7: return next device global rank, last device global rank

Table 2: Supported Sequence Length of Ring Attention and StarTrail on one Nvidia A100 80GB GPU.

| | | Supported Seq Len on one 80GB A100 GPU (K Tokens) | |
| --- | --- | --- | --- |
| Model Size | Length | Ring Attention | StarTrail |
| 3B | 128 | ✓ | ✓ |
| | 256 | ✓ | ✓ |
| | 512 | ✓ | ✓ |
| 7B | 128 | ✓ | ✓ |
| | 256 | ✓ | ✓ |
| | 512 | ✗ | ✗ |
| 13B | 128 | ✓ | ✓ |
| | 256 | ✗ | ✗ |
| | 512 | ✗ | ✗ |

StarTrail enhances the efficiency of the forward process with the help of torch JIT to fuse kernels aside from flash attention.

**Overlap communication with computing.** In StarTrail attention, P2P communication and self-attention computing are interleaved across iterations, each incurring considerable time. To mitigate this, StarTrail employs a double buffering technique to asynchronously execute communication and computing kernels, effectively overlapping these processes and enhancing GPU utilization.

**Save recomputation with checkpoints.** StarTrail adopts the checkpointing strategy introduced by DistFlashAttn[20], placing checkpoints at the end of the self-attention phase rather than the FFN of each transformer layer. This checkpoint placement effectively obviates the need to recompute the self-attention forward process during the backward pass, avoiding redundant attention computation.

# B  Additional Experiment

To comprehensively evaluate the memory consumption of StarTrail and Ring Attention, we compared the maximum supported sequence lengths of StarTrail with those reported in the Ring Attention paper [24]. As shown in Table 2, although StarTrail requires slightly more memory, it still supports sequence lengths commonly used in training tasks.

## B.1  Discussion

## B.2  Larger Batch Sizes and Models

We utilized small batches and model sizes in our experiments because these choices do not affect the underlying improvements in communication efficiency and computation-to-communication ratio that StarTrail provides. For larger batch sizes, both communication and computation scale proportionally, leaving the overlapping ability unchanged. Similarly, while larger models involve more layers or

larger hidden sizes, the key attention computations and corresponding ratios remain unaffected. Hence, our conclusions naturally extend to scenarios with larger batches and models.

### B.2.1 FlashAttention3 and Hopper GPUs

In addition to the original FlashAttention [8] used in our experiments, FlashAttention3 [35] has been introduced, specifically designed for Hopper and newer Nvidia GPUs. For FP16 precision, which is utilized in this paper, FlashAttention3 achieves a 1.5-2.0x speedup on Hopper GPUs. As discussed in Section 3, reducing attention computation overhead results in more P2P communication not being overlapped, further emphasizing the need to reduce communication volume. With the increasing adoption of Hopper GPUs, the significance of the StarTrail system will also grow.

### B.2.2 StarTrail and Other Parallelisms

**Model Parallelism** As is well known, **tensor parallelism** shards activations by attention heads during attention computation, making it easily combinable with StarTrail with minimal effort. However, when combined with tensor parallelism, the need for attention heads can limit the scalability of head-based sequence parallel methods like DeepSpeed-Ulysses. **Pipeline parallelism**, on the other hand, divides the model across layers without altering the computation patterns within Transformer blocks, making StarTrail orthogonal to it.

**Other Sequence Parallelism** StarTrail is orthogonal with other attention-head-sharding-based sequence parallelism approaches, such as DeepSpeed-Ulysses [14]. While DeepSpeed-Ulysses distributes attention heads across different devices, StarTrail can independently partition activations along the sequence length dimension. In future work, we can explore combining StarTrail with DeepSpeed-Ulysses to expand the communication scheduling space, harnessing the scalability of StarTrail alongside the efficiency of DeepSpeed-Ulysses.

In summary, StarTrail can be seamlessly integrated with other parallel training techniques, enabling the creation of a hybrid distributed training system.

## B.3 Other Related Works

**Attention Optimization**. Traditional full attention mechanisms necessitate $O(n^2)$ memory for storing the outputs of $QK^T$, leading to significant computational and memory demands. To address these challenges within the GPU, several approaches have been devised to reduce both memory and computational requirements. Memory-efficient attention[32] introduces a straightforward algorithm that requires only $O(1)$ memory relative to the sequence length, with an extension for self-attention that needs only $O(\log n)$ memory. FlashAttention further minimizes I/O overhead and enhances overall efficiency. Additionally, optimization methods specifically tailored for inference, such as PagedAttention[19], are also being developed to improve the efficiency of attention computations. In this work, we utilize FlashAttention within each iteration to reduce the computation overhead.

**Long-Sequence Training Techniques**. Sequence Parallelism[21] was initially introduced to enhance the efficiency of parallel long-sequence training. Ring Attention[24] improved communication efficiency through memory-efficient methods[32], supporting near-infinite sequence lengths. DeepSpeed-Ulysses[14] employs attention head splitting to achieve high efficiency, though it is constrained by the number of heads. Megatron Sequence Parallelism focuses on reducing memory costs during Tensor Parallelism, while DistFlashAttention[20] features a load-balance scheme and a novel gradient checkpoint method. Our work builds on these innovations, introducing a system that supports large-scale training with an efficient communication scheme.

**Techniques for Distributed Model Training**. Distributed model training encompasses two primary areas: 1) **Memory Management**: Various techniques aim to conserve GPU memory during distributed training, such as mixed precision training[26] and the ZeRO series[34]. In this work, we implement ZeRO-2 to manage optimizer states and gradients efficiently. 2) **Hybrid Parallelism**: Frameworks like Megatron[27] and Colossal AI[4] integrate multiple forms of parallelism. There are various existing Parallelism techniques like Pipeline Parallelism[13, 10, 23, 25] and Tensor Parallelism[37], which can be combined with StarTrail Parallelism to facilitate large-scale training. We are also considering the integration of additional frameworks such as [6] to enhance overlapping capabilities in future implementations.

