# OpenReview forum: "StarTrail: Concentric Ring Sequence Parallelism for Efficient Near-Infinite-Context Transformer Model Training"
_NeurIPS.cc/2025/Conference — NeurIPS 2025 poster_

### Official Review · Reviewer_nsiK · 2025-06-01

**Clarity:** 2
**Significance:** 3
**Originality:** 3
**Rating:** 4
**Confidence:** 3

**Summary:**

This paper proposes StarTrail, a new method to further reduce the communication overhead of training LLM with long sequences when the bandwidth between nodes and within nodes in a GPU cluster is different. Unlike Ring Attention, where each device processes only one input sequence block, StarTrail first groups multiple devices into a team to collaboratively process multiple sequence blocks, and then the intermediate results are aggregated to obtain the final result. The paper theoretically and empirically shows that StarTrail outperforms Ring Attention.

**Questions:**

The paper shows that StarTrail outperforms Ring Attention when there is a disparity between inter-node and intra-node bandwidths
in GPU clusters. However, how does StarTrail compare with SoTA methods like LoongTrain and USP which are also robust to network topologies?

**Ethical Concerns:**

["NO or VERY MINOR ethics concerns only"]

**Final Justification:**

After reviewing the authors' responses, I raised my final rating from 3 (Borderline reject) to 4 (Borderline accept). However, if I could, I would give it a 3.5 because the paper is poorly written in its current form. I don't think anyone who isn't an expert in the field would be able to understand it.

**Limitations:**

Yes.

**Paper Formatting Concerns:**

No.

**Quality:**

2

**Strengths And Weaknesses:**

Strengths:
* The paper studies an important problem.
* The paper proposes a sequence parallelism method that does not rely on Attention-Head-Sharding.

Weaknesses:
* The paper does not discuss some important related work. For example, LoongTrain [1] proposes Double-Ring-Attention, which partitions the GPUs into multiple inner rings, to fully utilize available NICs for inter-node communication.
* The theoretical analysis should not only compare StarTrail with Ring Attention but also include LoongTrain and Unified SP [2] (i.e., the combination of Ring Attention + DeepSpeed-Ulysses).
* The experiments should not only compare StarTrail with Ring Attention but also include LoongTrain and Unified SP (i.e., the combination of Ring Attention + DeepSpeed-Ulysses).
* Presentation could be improved.
  * The order of Figure 2 and Figure 3 is reversed.
  * There is no explanation for Figure 3 in the paper.
  * The notations used in Algorithm 1 are different from those listed in Figure 5, which makes it difficult for readers to understand.

[1] LoongTrain: Efficient Training of Long-Sequence LLMs with Head-Context Parallelism
[2] USP: A Unified Sequence Parallelism Approach for Long Context Generative AI

---

> ### Author Rebuttal · Authors · 2025-07-30
>
> Thanks for reading our paper. We would like to address your concerns as follows:
>
> 1. **About USP**. The main contribution of USP is to efficiently combine Ring Attention and Ulysses. As we discussed in Section 2.1.2, Ulysses is orthogonal to ring-style sequence parallelism, and our work specifically focuses on optimizing the performance of ring-style SP. As such, USP lies beyond the scope of this paper. Nevertheless, we are open to exploring potential combinations of StarTrail and Ulysses in future work on hybrid parallelism.
>
> 2. **About LoongTrain**. The most relevant aspect of LoongTrain to our work is its design of Double-Ring Attention. However, Double-Ring Attention pursues a fundamentally different objective. It partitions GPUs into $C$ outer rings and $P/C$ inner rings. The model first passes Keys and Values through each inner ring for one full round, followed by a single iteration through the outer ring. In this setup, the communication volume per inner ring round is $C2BNH$, repeated $\frac{P}{C}$ times, and the outer ring adds $\frac{P}{C}2BNH$. Thus, the total P2P communication volume becomes $2CBNH \times \frac{P}{C} + \frac{P}{C}2BNH = 2\frac{C+1}{C}PBNH$, which is slightly larger than the original Ring Attention ($2PBNH$). By contrast, StarTrail achieves a substantially smaller overall communication volume than Ring Attention. As stated in Section 4.5.2 of the LoongTrain paper, Double-Ring Attention improves computation-communication overlap **at the cost of slightly increased total communication volume**. StarTrail, on the other hand, simultaneously provides **lower communication volume**, **better overlap**, and **improved communication locality**, at the expense of a modest increase in memory usage. In summary, Double-Ring Attention explores a different tradeoff space in ring-style communication and **does not aim to reduce communication volume**. For this reason, we believe it is not appropriate to include it as a baseline, and a direct comparison would not be meaningful.
>
> 3. **Writing**. Thank you for your helpful suggestions. We will reverse the order of Figure 2 and Figure 3 and adopt clearer notations to enhance readability. Figure 3 serves as an overview of the entire system, with its components described in detail in Section 3. To further improve clarity, we will add an overview subsection following Section 3.1 to guide readers through the system structure in the revised version.
>
> Thanks again for your thoughtful feedback. We sincerely hope we have addressed your concerns.

---

> > ### Comment · Reviewer_nsiK · 2025-08-04
> >
> > Thank you for the reply.
> >
> > Based on your response, I think it would be great to use LoongTrain as a baseline since you claim that your method has many advantages over LoongTrain. Furthermore, LoongTrain is exploring different trade-offs of the ring attention as you said, aligning with the established principles of baseline selection in systems research.
> >
> > Therefore, I would not improve my score until you cover more baselines including LoogTrain.

---

> > > ### Author Response · Authors · 2025-08-05
> > > **Comparison results with LoongTrain**
> > >
> > > Thank you once again for your valuable feedback. To further substantiate our rebuttal, we have conducted additional experiments since over last response by implementing Double-Ring Attention ourselves, aiming to provide a fair and controlled comparison with StarTrail without interference from unrelated optimizations or external implementation differences.
> > >
> > > For a rigorous evaluation, we have carefully re-implemented Double-Ring Attention using the same infrastructure and methodology as StarTrail. This includes consistent use of both Torch and NCCL backends, fused computation kernels, activation checkpointing strategies, and other system-level details. All experiments were conducted on NVIDIA A100 and H100 GPUs, following the experimental setup described in Section 4 of our paper. The models used are identical to those reported in our main results.
> > >
> > > Moreover, we selected and tuned the best-performing configurations for both Double-Ring Attention and StarTrail to ensure a fair and meaningful comparison. The updated experimental results are presented below, and we hope these new findings help address your concerns more thoroughly.
> > >
> > > **Table R.1:**
> > >
> > > | Method | Model | GPU | Num. GPUs | Seq. Len. | Throughput (k tokens/s) |
> > > |--------|-------|-----|-----------|-----------|------------------------|
> > > | Ring Attention | DiT | A100_40GB_8 | 32 | 128k | 2.36 |
> > > | StarTrail | DiT | A100_40GB_8 | 32 | 128k | **5.06** |
> > > | Double-Ring Attention | DiT | A100_40GB_8 | 32 | 128k | 3.34 |
> > >
> > > **Table R.2:**
> > >
> > > | Method | Model | GPU | Num. GPUs | Seq. Len. | Throughput (k tokens/s) |
> > > |--------|-------|-----|-----------|-----------|------------------------|
> > > | Ring Attention | GPT | A100_40GB_16 | 32 | 128k | 3.71 |
> > > | StarTrail | GPT | A100_40GB_16 | 32 | 128k | **5.29** |
> > > | Double-Ring Attention | GPT | A100_40GB_16 | 32 | 128k | 4.12 |
> > >
> > > **Table R.3:**
> > >
> > > | Method | Model | GPU | Num. GPUs | Seq. Len. | Throughput (k tokens/s) |
> > > |--------|-------|-----|-----------|-----------|------------------------|
> > > | Ring Attention | GPT | H100_80GB_8 | 32 | 128k | 10.82 |
> > > | StarTrail | GPT | H100_80GB_8 | 32 | 128k | **14.57** |
> > > | Double-Ring Attention | GPT | H100_80GB_8 | 32 | 128k | 11.32 |
> > >
> > >
> > > For the three tables presented, we selected Double-Ring Attention configurations with inner-ring sizes of 2, 2, and 8, respectively, as these settings yielded the best performance. This selection aligns with the underlying hardware characteristics: NVIDIA H100 nodes are equipped with 8 × 400 Gbps NICs, enabling more efficient intra-node communication, while NVIDIA A100 nodes are limited to 100 Gbps Ethernet per node. The observed performance trends are also consistent with the discussion in Section 6.4 of the LoongTrain paper.
> > >
> > > As shown in the results, both StarTrail and Double-Ring Attention outperform baseline Ring Attention by improving the overlap between computation and communication. Notably, StarTrail achieves higher speedups, primarily due to its reduction in overall communication volume. The performance benefits of both methods are especially pronounced under weaker inter-node bandwidth conditions, where communication tends to be the bottleneck. Both StarTrail and Double-Ring Attention improve efficiency by enhancing communication-computation overlap, but through different trade-offs. StarTrail does so with slightly increased memory usage, while Double-Ring Attention incurs slightly higher communication volume.
> > >
> > > In summary, both StarTrail and Double-Ring Attention are effective approaches to improving the efficiency of Ring-style sequence parallelism. Double-Ring Attention provides moderate speedup while maintaining memory usage similar to Ring Attention, making it suitable for memory-constrained settings. StarTrail, on the other hand, delivers higher speedup by reducing communication volume, at the cost of slightly increased memory usage. These two methods offer complementary benefits and can be selected based on the specific system constraints and performance requirements—whether one prioritizes efficiency or memory footprint.

---

> > > > ### Comment · Reviewer_nsiK · 2025-08-05
> > > >
> > > > Thank you for your response. I adjusted my rating.
> > > >
> > > > Please include the comparison and discussion with LoongTrain in the next version of your paper. I believe this will improve the quality of your paper.

---

### Official Review · Reviewer_RamG · 2025-07-01

**Clarity:** 2
**Significance:** 2
**Originality:** 2
**Rating:** 4
**Confidence:** 2

**Summary:**

This paper introduces StarTrail, a new system for training Transformer models on very long sequences. The method improves upon existing ring-based sequence parallelism techniques. StarTrail organizes GPUs into teams and creates multiple smaller communication rings, called sub-rings. This design reduces the total peer-to-peer communication volume, especially across slower network links between compute nodes. The authors demonstrate that StarTrail achieves significant throughput gains over the standard Ring Attention method on both language and vision models. This performance improvement comes at the cost of a modest increase in GPU memory usage.

**Questions:**

Please see the weaknesses.

**Ethical Concerns:**

["NO or VERY MINOR ethics concerns only"]

**Final Justification:**

This paper presents significant throughput gains.

**Limitations:**

yes

**Paper Formatting Concerns:**

No major formatting concerns.

**Quality:**

2

**Strengths And Weaknesses:**

Strengths:

1. The core idea of creating hierarchical communication rings is a very effective solution to a clear problem. The paper correctly identifies that Ring Attention's single large ring is often bottlenecked by slow inter-node communication. By creating sub-rings that can operate within high-speed intra-node networks, the method directly addresses this critical performance issue.

2. The experimental evaluation is extensive and convincing. The authors test their method across multiple hardware types (A100, H100), different cluster topologies, and on two distinct Transformer architectures (GPT and DiT). This thorough testing on various setups provides strong evidence for the general applicability and robustness of their proposed system.

3. The paper presents a clear analysis of the performance trade-offs. The theoretical breakdown of communication and memory costs helps readers understand the fundamental principles behind StarTrail. It clearly explains how the team size parameter C allows a user to balance peer-to-peer communication, collective communication, and memory overhead to optimize for a specific hardware environment.

Weaknesses:

1. The comparison to prior work seems incomplete. The paper does not experimentally compare against other recent and relevant methods for long context training, such as DistFlashAttention.

2. The paper does not fully explore the scalability limitations of its own design. The All-Gather operation within each team could become a new bottleneck at a very large scale. The analysis does not discuss how the cost of this collective communication scales with the team size C or the total number of GPUs, nor does it address the C <= sqrt(P) constraint in depth.

3. The method's performance appears to be highly dependent on the network topology, but this dependency is not explored. The benefits of StarTrail rely on mapping teams and sub-rings to the physical hardware layout, for instance, keeping a sub-ring within a single node. The authors do not discuss how the system would perform on clusters with less favorable or more complex topologies where this mapping is not possible.

---

> ### Author Rebuttal · Authors · 2025-07-30
>
> Thanks for reading our paper! We would like to address your concerns as below:
>
> 1. **Related Work**. First, we would like to clarify that StarTrail focuses on optimizing the communication pattern of ring-style sequence parallelisms, and the currently most widely used baseline is Ring Attention. Other related works have focused on different aspects of sequence parallelism. For instance, we have discussed DistFlashAttention in Section B.3. DistFlashAttention proposes two main contributions. The first is a modification of the communication pattern to address the load imbalance issue in causal models. However, a more elegant solution, the Zigzag dataloader, has been proposed to address this problem. By simply modifying the data loading process, we can balance the load across GPUs. We have implemented the Zigzag dataloader in StarTrail, as discussed in Section A.1. The second contribution of DistFlashAttention involves reordering recomputation steps, which we have also implemented as a feature in StarTrail. StarTrail demonstrates further performance improvements by incorporating both of these techniques into the baseline. Therefore, we believe it is not necessary to include DistFlashAttention as an evaluation baseline. Nonetheless, we are happy to explore the integration of StarTrail with other optimization techniques in future work.
>
> 2. **Collective communication**. We would like to clarify that we have discussed how the cost of collective communication scales with the team size $C$ or the total number of GPUs in Section 3.2.2, line 221. As shown in Equation (2), the additional collective communication volume is much smaller than the reduced P2P communication volume, provided that $C<\sqrt{P}$. Moreover, we would like to emphasize that the condition $C<\sqrt{P}$ is a **feature**, rather than a limitation to be addressed. As discussed in Section 3.2.1, the valid range for $C$ is $1 < C < \sqrt{P}$. In our experiments, with $P=64$, $C$ should be greater than 1 and less than 8. When $C=1$, StarTrail degenerates to Ring Attention. When $C=8$, there are no rings; each GPU in a team first gathers Keys and Values from the other 7 teams, performs attention computation, and applies reduce-scatter to obtain the final result. The optimal choice of $C$ depends on the system configuration, and StarTrail achieves the best performance at the sweet spot that balances reduced P2P communication and overlapping collective operations with matrix multiplications.
>
> 3. **Topology of hardware**. We would like to clarify that our evaluation includes four different cluster topologies to demonstrate that StarTrail consistently outperforms the baseline regardless of hardware configuration. As shown in Table 1, our experiments span 2 GPU types, 4 topologies, and 2 types of inter-node connections, covering most common configurations used in modern Transformer training workloads. We would also like to emphasize that StarTrail does not rely on keeping sub-rings within a single node to achieve performance gains. We consider two possible scenarios, and we refer to inter-node bandwidth as $W_l$ and intra-node as $W_h$. In the first, all ring-style communication occurs within nodes. In this case, the communication overhead is $(\frac{P}{C^2}-1)(\frac{2CBNH}{PW_h} + L)=\frac{(P-C^2)2BNH}{CPW_h} + (\frac{P}{C^2}-1)L$. In the second scenario, where sub-ring communication spans multiple nodes, the communication is bottlenecked by inter-node bandwidth, yielding $(\frac{P}{C^2}-1)(\frac{2CBNH}{PW_l} + L)=\frac{(P-C^2)2BNH}{CPW_l} + (\frac{P}{C^2}-1)L$. As discussed in the paper, StarTrail reduces communication overhead by approximately $\frac{W_h}{W_l}C$ and $C$ in these respective cases. As for less favorable environments, it becomes even more difficult to overlap communication with computation, thus making StarTrail more beneficial by reducing communication volume. For more complex topologies, communication is still constrained by the lowest-bandwidth links in the ring, and the corresponding theoretical analysis remains similar. We would also like to clarify that our goal is to address challenges in common long-sequence training settings, while more exotic or complex cluster topologies are beyond the scope of this work.
>
> Thanks again for your suggestions, and we sincerely hope we have addressed your concerns.

---

> > ### Author Response · Authors · 2025-08-05
> > **Additional Baseline**
> >
> > Thank you once again for your valuable feedback. To further substantiate our rebuttal, we have conducted additional experiments since over last response by implementing the Double-Ring Attention from LoongTrain (as required by Reviewer nsiK) ourselves, aiming to provide a fair and controlled comparison with StarTrail without interference from unrelated optimizations or external implementation differences.
> >
> > For a rigorous evaluation, we have carefully re-implemented Double-Ring Attention using the same infrastructure and methodology as StarTrail. This includes consistent use of both Torch and NCCL backends, fused computation kernels, activation checkpointing strategies, and other system-level details. All experiments were conducted on NVIDIA A100 and H100 GPUs, following the experimental setup described in Section 4 of our paper. The models used are identical to those reported in our main results.
> >
> > Moreover, we selected and tuned the best-performing configurations for both Double-Ring Attention and StarTrail to ensure a fair and meaningful comparison. The updated experimental results are presented below, and we hope these new findings help address your concerns more thoroughly.
> >
> > **Table R.1:**
> >
> > | Method | Model | GPU | Num. GPUs | Seq. Len. | Throughput (k tokens/s) |
> > |--------|-------|-----|-----------|-----------|------------------------|
> > | Ring Attention | DiT | A100_40GB_8 | 32 | 128k | 2.36 |
> > | StarTrail | DiT | A100_40GB_8 | 32 | 128k | **5.06** |
> > | Double-Ring Attention | DiT | A100_40GB_8 | 32 | 128k | 3.34 |
> >
> > **Table R.2:**
> >
> > | Method | Model | GPU | Num. GPUs | Seq. Len. | Throughput (k tokens/s) |
> > |--------|-------|-----|-----------|-----------|------------------------|
> > | Ring Attention | GPT | A100_40GB_16 | 32 | 128k | 3.71 |
> > | StarTrail | GPT | A100_40GB_16 | 32 | 128k | **5.29** |
> > | Double-Ring Attention | GPT | A100_40GB_16 | 32 | 128k | 4.12 |
> >
> > **Table R.3:**
> >
> > | Method | Model | GPU | Num. GPUs | Seq. Len. | Throughput (k tokens/s) |
> > |--------|-------|-----|-----------|-----------|------------------------|
> > | Ring Attention | GPT | H100_80GB_8 | 32 | 128k | 10.82 |
> > | StarTrail | GPT | H100_80GB_8 | 32 | 128k | **14.57** |
> > | Double-Ring Attention | GPT | H100_80GB_8 | 32 | 128k | 11.32 |
> >
> >
> > For the three tables presented, we selected Double-Ring Attention configurations with inner-ring sizes of 2, 2, and 8, respectively, as these settings yielded the best performance. This selection aligns with the underlying hardware characteristics: NVIDIA H100 nodes are equipped with 8 × 400 Gbps NICs, enabling more efficient intra-node communication, while NVIDIA A100 nodes are limited to 100 Gbps Ethernet per node. The observed performance trends are also consistent with the discussion in Section 6.4 of the LoongTrain paper.
> >
> > As shown in the results, both StarTrail and Double-Ring Attention outperform baseline Ring Attention by improving the overlap between computation and communication. Notably, StarTrail achieves higher speedups, primarily due to its reduction in overall communication volume. The performance benefits of both methods are especially pronounced under weaker inter-node bandwidth conditions, where communication tends to be the bottleneck and opportunities for overlap are more limited. Both StarTrail and Double-Ring Attention improve efficiency by enhancing communication-computation overlap, but through different trade-offs. StarTrail does so with slightly increased memory usage, while Double-Ring Attention incurs slightly higher communication volume.
> >
> > In summary, both StarTrail and Double-Ring Attention are effective approaches to improving the efficiency of Ring-style sequence parallelism. Double-Ring Attention provides moderate speedup while maintaining memory usage similar to Ring Attention, making it suitable for memory-constrained settings. StarTrail, on the other hand, delivers higher speedup by reducing communication volume, at the cost of slightly increased memory usage. These two methods offer complementary benefits and can be selected based on the specific system constraints and performance requirements—whether one prioritizes efficiency or memory footprint.
> >
> > If you have any remaining concerns, please kindly let us know. Thanks!

---

> > > ### Comment · Reviewer_RamG · 2025-08-05
> > > **Thanks for your response.**
> > >
> > > Thanks for your response. Most of my concerns are solved.

---

### Official Review · Reviewer_xk9c · 2025-07-02

**Clarity:** 2
**Significance:** 3
**Originality:** 3
**Rating:** 3
**Confidence:** 4

**Summary:**

In this work, a novel sequence/context parallelism strategy called StarTrail is proposed. By introducing a new parallelism dimension, ring communication can be split into subrings, thereby reducing communication volume. This is particularly beneficial when inter-node communication is a bottleneck due to limited bandwidth. Evaluation results show that StarTrail significantly outperforms the original Ring Attention, achieving higher throughput across various settings.

**Questions:**

1. How does inter-node bandwidth affect performance? Intuitively, with higher bandwidth, the improvement might be limited since communication is no longer the bottleneck - especially if communication can be overlapped with computation. Some discussion or insight into the relationship between bandwidth and exposed communication would be helpful.

2. Does the evaluation assume pure context parallelism (i.e., CP degree equal to the number of GPUs)? If so, this may not be a practical setup and could result in a weaker baseline, as hybrid parallelism generally yields better throughput or MFU. For example, the open-source training framework NeMo provides a training recipe for LLaMA3-128k [1], where the CP degree is only 8. A total CP degree much higher than 8 may not be realistic.

3. In Figure 8, why is the rightmost subfigure normalized to 0.5 instead of 1?

[1] http://github.com/NVIDIA/NeMo/blob/main/nemo/collections/llm/recipes/llama3_8b_128k.py

**Ethical Concerns:**

["NO or VERY MINOR ethics concerns only"]

**Final Justification:**

Although the authors put significant effort into the rebuttal, the paper still suffers from several issues, including insufficient discussion and evaluation, poor readability, and unclear implementation. For these reasons, I will maintain my score.

**Limitations:**

See Weaknesses and Questions.

**Quality:**

2

**Strengths And Weaknesses:**

1. The motivation and problem setting are reasonable: inter-node bandwidth is clearly a bottleneck for Ring Attention.

2. The analysis and evaluation look promising.

3. Inter-node and intra-node bandwidths differ significantly and can have distinct impacts. The analysis assumes a single bandwidth value W, which may undermine the correctness of the conclusions.

4. The terminology is inconsistent: Team group, ring communication group, and subring appear to refer to the same concept but are used interchangeably.

5. Only values of C = 2 and C = 4 are discussed. With only two data points, it's difficult to assess the full impact of C. Including results for C = 8 (especially for 64 or more GPUs) would be helpful. Additionally, further discussion on why C = 2 or C = 4 performs better across different settings would improve understanding.

6. There is no discussion on convergence. Including a training curve comparison would provide the most straightforward evidence of the correctness of the proposed algorithm.

7. The topology hierarchy is unclear. Based on my understanding, the hierarchy and corresponding communication are as follows:

    Team - C GPUs - all-gather

    Team group - P / C² teams - subring communication

    What is the communication pattern between the C team groups?

---

> ### Author Rebuttal · Authors · 2025-07-30
>
> Thanks for reading our paper and providing constructive suggestions! We sincerely appreciate your feedback and would like to address your concerns as follows.
>
> 1. **Bandwidth in the analysis**. In our theoretical analysis, we assumed a single bandwidth $W$ to simplify the presentation. However, we are happy to extend the analysis to incorporate both inter-node and intra-node bandwidths. Specifically, we denote the inter-node bandwidth as $ W_l $ and intra-node bandwidth as $W_h$. For Ring Attention, as illustrated in Figure 2, the overall communication bandwidth is constrained by the bottleneck $W_l$, and the communication overhead can be expressed as $(P - 1)(\frac{2BNH}{PW_l} + L) = \frac{2BNH(P-1)}{W_lP} + (P - 1)L$.
>
>    For StarTrail, we consider two possible scenarios. In the first case, all ring-style communication occurs within nodes. The corresponding communication overhead is $(\frac{P}{C^2}-1)(\frac{2CBNH}{PW_h} + L)=\frac{(P-C^2)2BNH}{CPW_h} + (\frac{P}{C^2}-1)L$. In the second case, where sub-ring communications span across nodes, the bandwidth is again limited by $W_l$, leading to a communication cost of $(\frac{P}{C^2}-1)(\frac{2CBNH}{PW_l} + L)=\frac{(P-C^2)2BNH}{CPW_l} + (\frac{P}{C^2}-1)L$. As in our main analysis, StarTrail reduces communication overhead by approximately $\frac{W_h}{W_l}C$ and $C$ in the two cases, respectively.
>
> 2. **Topology and related communication**. We would like to clarify several key concepts introduced in the paper:
>    - **Teams**: Groups of $C$ GPUs that perform all-gather and reduce-scatter operations; for example, 2 GPUs per team as shown in Figure 3(b).
>    - **Subrings**: Smaller rings used for ring-style communication, each comprising $P/C^2$ GPUs (e.g., 4 GPUs in Figure 3(b)).
>    - **Team groups**: Collections of subrings that are connected via all-gather and reduce-scatter. In Figure 3(b), there are two team groups: the first includes teams 0–3 and the second includes teams 4–7. Communication between team groups only occurs during the initial P2P communication phase (Figure 6, Section A.2). In subsequent iterations of ring-style and collective communication, no inter-group communication occurs.
>
> 3. **The value of C**. As detailed in Section 3.2.1, the parameter $C$ satisfies $1 < C < \sqrt{P}$. In our experiments with $P=64$, valid values for $C$ are in the range (1, 8). When $C = 1$, StarTrail reduces to Ring Attention. When $C = 8$, rings are eliminated entirely: each GPU gathers Keys and Values from the other 7 teams, performs attention computation, and applies reduce-scatter for final aggregation. This scenario introduces large-scale collective communications that cannot be overlapped with matrix multiplications, leading to performance degradation—hence we did not include it in our main figures. However, we are happy to include this case in the appendix of the revised version. The optimal choice of $C$ depends on system characteristics, and StarTrail achieves the best performance at the sweet point of the tradeoff between the reduction of P2P communication and the collective operations, which can be partially overlapped with matmuls.
>
> 4. **Convergence**. StarTrail is mathematically equivalent to FlashAttention (and thus to Ring Attention). As described in Algorithm 1, StarTrail computes attention scores block by block, preserving intermediate values $lse$ and $O$ and updating them iteratively. The only difference is that StarTrail computes $C$ intermediate scores in parallel and merges them using reduce-scatter, rather than updating scores sequentially as in FlashAttention. These approaches are mathematically equivalent. Therefore, StarTrail maintains the same convergence behavior as Ring Attention while improving runtime. That said, we are willing to provide convergence curves in future versions for completeness.
>
> 5. **Hybrid Parallelism**. We address this concern from three perspectives.
>    First, sequence parallelism (particularly Ring-style) operates independently of other forms like tensor or pipeline parallelism and is typically deployed across nodes. This placement makes it more challenging to overlap communication with computation, which further motivates the need for StarTrail.
>    Second, emerging workloads—such as multi-modal models or long chain-of-thought reasoning—require sequence lengths far exceeding 128k tokens, necessitating higher degrees of sequence parallelism.
>    Third, even at small degrees of CP(SP) (e.g., 8), StarTrail delivers improved performance with small team sizes (e.g., 2), as demonstrated in Figure 10(a) and (b). Thus, we believe StarTrail is necessary to support the scaling of modern Transformer models with increasingly longer sequences.
>
> 6. **Bandwidth in evaluations**. The benefit of StarTrail is related to the communication bandwidth: the lower the bandwidth, the more significant the advantage. So the necessity of optimizing the communication can be weakened if the communication can be well-covered by computation. However, as we have discussed in the last paragraph, sequence parallelisms are usually placed between nodes in real scenarios so that parallelisms that require more communication, like tensor parallelism, can be kept within the nodes. This makes it hard for CP(SP) to be overlapped. To evaluate bandwidth impact, we included two types of interconnects: $8 \times 400$ Gbps InfiniBand (which is among the highest inter-node connection bandwidths and typically deployed on the newest NVIDIA DGX systems) and 100 Gbps Ethernet (common in cloud A100 setups), as shown in Table 1. Experimental results show that StarTrail significantly outperforms Ring Attention even under high-bandwidth settings like 8*400 Gbps InfiniBand, underscoring its practical effectiveness.
>
> 7. **Typo**. We apologize for the typo in Figure 8. The y-axis should indeed be normalized to 1. We will correct this in the revised version.
>
> Thanks again for your thoughtful suggestions and careful reading of our paper. We sincerely hope that our responses have addressed your concerns.

---

> > ### Author Response · Authors · 2025-08-06
> > **Any remaining concerns?**
> >
> > Dear Reviewer,
> >
> > I hope this message finds you well. As the discussion period is near its end, with less than three days left, we sincerely hope to ensure that we have addressed all your concerns. If there are any remaining points you would like to discuss, please kindly let us know. Your insights are valuable to us, and we are looking forward to your reply.
> >
> > Thanks again for reading our paper and providing such thoughtful suggestions.

---

> > ### Comment · Reviewer_xk9c · 2025-08-06
> >
> > Thank you for the detailed responses. I have some follow-up questions:
> >
> > 1. Can you report the intra-node bandwidth and highlight the inter/intra-node bandwidth ratio in Table 1?
> >
> > 2. In both the paper and the rebuttal, you emphasize that in some cases, computation and communication can be easily overlapped, while in others they cannot, or only partially. However, it's difficult for readers to assess these claims. Can you provide quantitative evidence showing how much overlap is actually achieved?
> >
> > 3. The theoretical communication cost can be significantly reduced, but this improvement is not reflected in the throughput. How do you explain this gap, and how can the formulation be improved to better align with the evaluation results?
> >
> > 4. What framework do you use for the Ring Attention baseline? How is your algorithm implemented? This information should be included for reproducibility.
> >
> > 5. For GPT-7B with a sequence length of 64k, the model FLOPs is roughly 9400 TFLOPs (based on HuggingFace Transformer Flops Calculator).
> > In Figure 7, the throughput is approximately 35k tokens/s.
> > Therefore, the FLOPS per H100 GPU is: 9400 / (64/35) / 32 ≈ 160 TFLOPS.
> >
> >     The peak performance of an H100 with BF16 is 989 TFLOPS, so the model FLOPs utilization is: 160/989 ≈ 16%
> >
> >     Please let me know if this calculation is correct.

---

> > > ### Author Response · Authors · 2025-08-07
> > > **Response to follow-up questions**
> > >
> > > Thanks for your response, and we would like to answer the new questions as follows:
> > >
> > > 1. In our experiments, we used two types of GPUs. The first one is the Nvidia A100 40GB, which provides 300 GB/s = 2400 Gbps NVLink (each direction) and a 100Gbps Ethernet inter-node connection shared by 8 GPUs (12.5Gbps per GPU). The second one is the Nvidia H100 80GB, with 450 GB/s = 3600 Gbps NVLink (each direction) and 8×400Gbps InfiniBand inter-node connections (400Gbps per GPU). Therefore, the inter/intra-node bandwidth ratios for the two types of GPUs are 12.5/2400 = 0.0052 and 400/3600 = 0.111, respectively.
> > >
> > > 2. We are glad to report the actual overlap between communication and computation achieved. We believe the most effective way to demonstrate this is to provide the actual Torch profiler trace files. However, as it is prohibited to provide extra images or files during the review process, we present selected data points that approximate the upper bound of the overlap. We used Torch profiler to trace Ring Attention and recorded the communication and computation overhead as follows:
> > >
> > > | GPU | num nodes | num GPUs per node | inter-node bandwidth | Seq. Len. | Fwd. Comm. (ms) | Fwd. Comp. (ms) | Bwd. Comm. (ms) | Bwd. Comp. (ms) |
> > > | --- | --- | --- | --- | --- | --- | --- | --- | --- |
> > > | H100 | 4 | 8 | 8\*400Gbps | 64k | 603 | 175.8 | 1460.4 | 516.3 |
> > > | H100 | 4 | 8 | 8\*400Gbps | 128k | 1246 | 700.1 | 3279.9 | 2076.4 |
> > >
> > > We used input with num_heads = 32 and head_dim = 128. Two key observations are: a) Communication overhead is larger than computation overhead, especially for shorter sequences; b) Although we cannot show the trace in the rebuttal, we observed that the computation kernel is not always launched concurrently with the communication kernel, meaning that the actual overlap is smaller than the theoretical upper bound. It is also worth noting that the 8×400Gbps inter-node bandwidth setup already represents one of the most performant GPU clusters. On more cost-effective clusters using Ethernet, communication overhead would be even higher.
> > >
> > > 3. We believe the gap between the theoretical communication volume and the observed end-to-end throughput stems from the following factors: a) As mentioned above, part of the communication can still be overlapped with the attention computation; b) The reported end-to-end throughput includes other components of training, such as MLP computation and communication overhead from the ZERO-2 optimizer state and gradient sharding. Reporting only the attention module would show a higher "speedup ratio," but we chose to report end-to-end throughput for a more comprehensive evaluation; c) As discussed in Point 3 of our previous response, a higher value of C does not always yield better performance. Although the analytical formulation can be refined by incorporating Equation 2, the collective communication described there is also partially overlapped with matmul operations. Hence, we argue that the most efficient and accurate approach is to run profiling once before training, which can then be reused across the same environment.
> > >
> > > 4. Thank you for the reminder. Some implementation details are available in Section A.3 of the appendix. To implement StarTrail and Ring Attention, we built upon the publicly available Ring-Flash-Attention GitHub code, with minimal modifications for StarTrail. For other operations, we adopted Torch-native code as much as possible to avoid any confounding optimizations from frameworks like Megatron. We also ensured that the implementations of StarTrail and Ring Attention are identical except for the ring-style communication logic.
> > >
> > > 5. The MFU calculation is correct according to the data. We believe the relatively low MFU is due to two primary reasons: a) Owing to the nature of Ring-style Sequence Parallelism, the heavy communication cost (even with StarTrail) renders the attention process communication-bound, leading to GPU idling during communication; b) As mentioned above, we intentionally avoided using end-to-end training frameworks in order to maintain a clean and fair comparison between the baseline and our method. The MFU could be improved by incorporating additional optimization techniques (not related to the scope of this work), such as a more optimized implementation of ZERO-2.
> > >
> > > Thanks again for your insightful comments and suggestions. We hope our response adequately addresses your concerns.

---

> > > > ### Comment · Reviewer_xk9c · 2025-08-08
> > > >
> > > > Thank you again for the detailed responses. I have some follow-up questions:
> > > >
> > > > 1.	According to your first response, StarTrail reduces communication overhead by approximately W_h / W_l × C. Does this mean the communication overhead can be reduced by 2400/12.5×4=768x?
> > > > Also, given that the inter-/intra-node bandwidth ratios vary significantly between A100 and H100 systems (0.0052 vs. 0.111), yet the speedups remain similar, does this imply that inter-node bandwidth is not strongly correlated with the speedup? More specifically, how does inter-node bandwidth affect speedup?
> > > >
> > > > 2.	Regarding the table you provided, unfortunately I couldn’t find any information related to overlap.
> > > > By the way, I’m unsure why the backward communication time is roughly 2.5× the forward communication time, and why the backward computation time is roughly 3× the forward computation time.
> > > >
> > > > 3.	> The reported end-to-end throughput includes other components of training, such as MLP computation and communication overhead from the ZERO-2 optimizer state and gradient sharding.
> > > >
> > > >     It might be better to conduct an ablation study focused on the attention module only to more clearly isolate its contribution.
> > > >
> > > > Other issues:
> > > >
> > > > 4.	In Table 1, you describe a system with 8 nodes and 4 devices per node. However, the evaluations don’t seem to use this system. Where exactly was this system used, and what are the corresponding results?
> > > >
> > > > 5.	In the strong and weak scaling experiments (Section 4.3), only the number of GPUs is mentioned. Since different system topologies can significantly affect performance, I believe the specific system used should be clearly stated.

---

> > > > > ### Author Response · Authors · 2025-08-08
> > > > >
> > > > > Thanks for reading our response. We would like to answer your follow-up questions as follows:
> > > > >
> > > > > 1. We would like to clarify that our provided calculations are based on theoretical analysis, which should be considered as an upper bound of the real-world speedup. In actual experiments, the observed speedup is influenced by various factors, such as the current network traffic of the entire cluster, the design of the NCCL communication kernel, and other system-level conditions. That said, we are confident that a higher $W_h / W_l$ ratio leads to higher achievable speedup, which is empirically supported by the four sets of scaling experiments shown in Figures 9 and 10. Specifically, Figures 9(a) and 10(a), which use A100 clusters with 100Gbps inter-node connections, demonstrate higher speedup than Figures 9(b) and 10(b), which use H100 clusters with 3200Gbps inter-node bandwidth. We emphasize that the theoretical analysis serves to help readers better understand the source of our speedup, rather than to predict precise performance (which depends on many practical factors). The ultimate goal of this work is to present an efficient and practical technique for long-sequence training, which we validate in the experimental sections.
> > > > >
> > > > > 2. (a) The overlap in ring-style attention refers to two components: overlapping forward communication with forward computation, and overlapping backward communication with backward computation. From the table, the overlap ratio can be calculated as Comp./Comm.       (b) The communication time during the backward pass is approximately 2x the forward pass because, during forward propagation, only Keys and Values are communicated. In contrast, during backward propagation, Queries, Outputs, Gradients of Queries, and Gradients of Outputs are all transmitted. As a result, the communication volume during backward propagation is twice that of the forward pass. Moreover, the computation workload is also heavier during backward propagation due to the calculation of gradients. The actual overhead difference depends on implementation details. Please refer to the code of flashattention for more details.
> > > > >
> > > > > 3. We need to emphasize that the rest of the computations and communications in the reported end-to-end speedup are exactly the same for our method and the baselines. The end-to-end speedup we report can be represented as:  $$\frac{T_{ring} + T_{others}}{T_{star} + T_{others}}$$ , while the attention speedup is $$\frac{T_{ring}}{T_{star}}$$ . Therefore, the attention-specific speedup of StarTrail is guaranteed to be higher than the overall end-to-end speedup we report. While we would like to include this ablation study in our paper, we are unfortunately constrained by time, as there are less than 24 hours remaining in the discussion period. We plan to add this ablation in a future version, as it further clarifies StarTrail's performance but does not change the paper’s core conclusions.
> > > > >
> > > > > 4. Sorry, we originally intended to conduct experiments on this cluster, but later decided to exclude it due to the rare usage of such a topology and forgot to remove it from the table. We will remove the reference to this cluster from our paper.
> > > > >
> > > > > 5. The types of GPUs are specified in the captions of Figures 9 and 10. For both Nvidia A100 and H100 experiments, we used clusters with 8 GPUs per node, with bandwidths as described in Table 1. Thank you for the reminder—we will incorporate these clarifications into the paper.
> > > > >
> > > > > We sincerely hope our response can address your concerns.

---

> > > > > > ### Comment · Reviewer_xk9c · 2025-08-09
> > > > > >
> > > > > > I don’t think using Figures 9 and 10 to claim that a system with lower inter-node bandwidth can achieve higher speedup makes sense, since the models are different. If we use the two subfigures at the bottom right of Figure 7, we would reach the opposite conclusion—that the H100 machine (with higher inter-node bandwidth) can achieve higher speedup.
> > > > > >
> > > > > > Overall, I appreciate the authors’ effort during the rebuttal. Given that:
> > > > > >
> > > > > > 1. Important information and discussions are missing in the main text, such as the impact of intra-/inter-node bandwidth and the computation–communication overlap ratio, and it seems difficult to incorporate the new content provided during the rebuttal period into the paper.
> > > > > >
> > > > > > 2. There are too many incorrect tables and figures:
> > > > > >     * Table 1 includes a system that was not used for evaluation, and the caption mentions “#GPU” but there is no such column.
> > > > > >     * Figure 5 is actually a table, and there is no reference to it in the main text.
> > > > > >     * In Figure 7, some units on the y-axis are missing.
> > > > > >     * Figure 8 contains incorrect normalization.
> > > > > >
> > > > > > 3. There are too many grammatical and formatting issues:
> > > > > >     * L293: “StarTrail’s P2P communication is mostly confined intra-node”
> > > > > >     * L312: “As is depicted”
> > > > > >     * L313: “can also explained by”
> > > > > >     * L315: “as explained the previous sections”
> > > > > >     * L289: “As indicated in the figure, in all six settings” should be “eight settings”
> > > > > >     * L291: “2.114x, 1.414x, 1.629x, 1.425x, 1.360x, 1.199x, 1.771x, and 1.346x” — these numbers are not very informative, as it is unclear which number corresponds to which experiment.
> > > > > >     * When referring to a figure or table, capitalization is inconsistent.
> > > > > >     * Sometimes there is no space between a number and its unit, between a word and a parenthesis, or between a word and a reference number.
> > > > > >
> > > > > > I believe it would be challenging to improve the quality of this manuscript with only minor revisions. Therefore, I maintain my original score. However, I remain open to discussion with other reviewers and the AC.
> > > > > >
> > > > > > Furthermore, I suggest that the authors consider merging Figures 3, 4, and 6 to free up space for further discussion.

---

> > > > > > > ### Author Response · Authors · 2025-08-09
> > > > > > >
> > > > > > > We sincerely appreciate the time and effort you have dedicated to reviewing our work and engaging in the discussion during the rebuttal period.
> > > > > > >
> > > > > > > As stated clearly in Section 1, the central claim of our paper is: “With very little additional memory cost, StarTrail parallelism significantly reduces the peer-to-peer communication volume.” Our extensive experimental results across multiple settings consistently validate this claim, showing that StarTrail achieves substantial speedups and efficiency gains in long-sequence training. These results directly address a long-standing bottleneck in distributed training for large models.
> > > > > > >
> > > > > > > We acknowledge the minor presentation and formatting issues you noted, and we are confident these can be quickly resolved without altering the substance of our work. Such small editorial matters should not overshadow the demonstrated technical merit, novelty, and practical impact of the proposed method.
> > > > > > >
> > > > > > > Given the strength of the empirical evidence and the significance of the problem addressed, we respectfully urge you to reconsider your decision. We believe StarTrail offers a valuable contribution that can benefit the community and foster future advances in efficient long-sequence model training.

---

> > > ### Author Response · Authors · 2025-08-08
> > >
> > > Dear reviewers,
> > >
> > > Sorry for sending another message. As there is approximately only one day left for the discussion period, we would appreciate it if you could left us know if our last response has addresses your follow up questions. Thanks again for your great efforts in assisting us to improve our work!
> > >
> > > Best regards,
> > > Authors

---

> ### Comment · Reviewer_nsiK · 2025-08-07
>
> Thank you for the comments from Reviewer xk9c and also author.
>
> After reading your discussion, the discussion from my side, and also the LoongTrain paper, I have a new question.
> * According to the HuggingFace Transformer Flops Calculator and Figure 7 of your paper, I find that for 128k sequence, StarTrail can achieve **MFU ≈ 12%**
> * However, under the same setting, the LoongTrain paper shows that they can achieve nearly **MFU ≈ 40%** as shown in Figure 14 of their arxiv paper.
> * Meanwhile, your implementation of LoongTrain is 20% slower than StarTrail.
>
> What are your thoughts on the numbers?

---

> > ### Author Response · Authors · 2025-08-07
> > **To Reviewer nsiK**
> >
> > Thank you for your response and the opportunity to clarify some important points regarding the MFU and the implementation details of LoongTrain.
> >
> > 1. **Clarification on MFU Reporting in LoongTrain**:
> > The 40% MFU reported in Figure 14 of the LoongTrain paper refers specifically to 2D Attention, which combines Ulysses and Double Ring Attention, rather than to Double Ring Attention alone. This distinction is crucial. Due to its all-to-all communication pattern, Ulysses inherently achieves higher MFU than Ring-style attention mechanisms, although its scalability is constrained by the number of attention heads. In contrast, the Double Ring Attention we implemented in our rebuttal is a pure version of Double Ring Attention without Ulysses integration, to ensure a clean and fair comparison with our proposed method. We believe this is the main reason for the MFU discrepancy between our results and the ones reported in LoongTrain.
> >
> > 2. **Consistency in Implementation Framework**:
> > As noted in Point 5 of our “Response to Follow-up Questions” for Reviewer xk9c, we deliberately avoided integrating our implementation into end-to-end training frameworks. This decision was made to eliminate confounding variables and ensure that both our proposed method and the baselines—including Double Ring Attention—are compared within a consistent and controlled environment. Meanwhile, the LoongTrain paper reports MFU results from its integration into InternEvo, a highly optimized end-to-end training system. In contrast, we implemented Double Ring Attention using the same codebase and settings as StarTrail, again for fairness. Now that StarTrail has demonstrated clear efficiency gains over baselines, we are also planning to explore its integration into optimized frameworks such as Megatron in future work.
> >
> > We hope these clarifications help contextualize our results and highlight our efforts to maintain a fair and rigorous evaluation.

---

### Official Review · Reviewer_LSQf · 2025-07-04

**Clarity:** 2
**Significance:** 2
**Originality:** 3
**Rating:** 5
**Confidence:** 2

**Summary:**

This paper proposes a training system named StarTrail, which uses a concentric ring parallelism scheme to train Transformer models on near-infinite-context sequences. Specifically, the method introduces an additional parallel dimension by grouping GPUs into teams and divides the communication workload into multiple sub-rings. This approach reduces the total peer-to-peer communication volume and avoids bottlenecks caused by slower inter-node connections. The authors conduct experiments on both Natural Language Processing (GPT) and Computer Vision (DiT) models using various hardware clusters equipped with Nvidia A100 and H100 GPUs. The experiments show that StarTrail achieves performance improvements over existing methods without affecting the computational results.

**Questions:**

N/A

**Ethical Concerns:**

["NO or VERY MINOR ethics concerns only"]

**Final Justification:**

Per previous discussion: I maintain my original score of 5 - Accept.

**Limitations:**

yes

**Quality:**

3

**Strengths And Weaknesses:**

**Strength**

The proposed method improves training throughput for long-sequence Transformer models. By dividing communication into concentric sub-rings, it reduces P2P communication volume and achieves performance speedups.

**Weaknesses**

1. The primary weakness in terms of significance is that StarTrail is an incremental improvement upon a specific existing method, Ring Attention.  It is a systems-level optimization that re-architects the communication pattern rather than a fundamental breakthrough in attention algorithms.Its scope is confined to this specific paradigm.

2. The core idea involves dividing GPUs into teams and forming sub-rings. While Figures 4 and 6 are helpful, the description of the initialization process and the mapping of ranks to sub-rings and teams can be difficult to follow on a first read. The logic is spread between the main text, figures, and algorithms in the appendix, which could be integrated more smoothly to improve comprehension.

---

> ### Author Rebuttal · Authors · 2025-07-30
>
> Thank you very much for your thoughtful and constructive feedback, and for recognizing the contributions of our work.
>
> StarTrail is a system-level optimization designed for long-sequence Transformer training. It modifies the order of computation while preserving equivalence with the original attention mechanism. As the paradigm of sequence parallelism has largely converged to Ulysses-style and Ring-style Attention, we focus in this work on optimizing the performance of the Ring-style variant, which is widely applicable in practice. That said, we are also interested in extending StarTrail to other attention mechanisms, such as Linear Attention, and appreciate the suggestion to consider broader applicability.
>
> We acknowledge that the hierarchical structure of StarTrail may be challenging to follow upon first reading. In response to your comments, we will revise the manuscript to provide clearer and more detailed explanations of the process, and will move essential content from the appendix into the main text to enhance clarity.
>
> Once again, thank you for your valuable insights and for helping us improve our work.

---

> > ### Comment · Reviewer_LSQf · 2025-08-06
> >
> > Thanks for your response! I maintain my original score of 5 - Accept.

---

### Note · Authors · 2025-08-14

We would like to express our sincere appreciation to the reviewers and chairs for their valuable time and effort in reviewing our work. We greatly enjoyed the discussions during the rebuttal period and have made every effort to address the concerns raised. The feedback and exchanges have been instrumental in improving our paper.

In this work, we propose StarTrail, a novel diagram for ring-style sequence parallelism designed to improve long-sequence training efficiency by reducing peer-to-peer communication volume. We have conducted extensive experiments across different models and computing clusters to evaluate the effectiveness of the proposed technique.

We sincerely hope that StarTrail can offer new insights and make meaningful contributions to the community.

---

### Decision · Program_Chairs · 2025-09-17

**Decision:**

Accept (poster)

**Comment:**

The authors propose StarTrail, a distributed training system for training Transformers on long sequences. Traditional data paradigms like data parallelism, tensor parallelism and pipeline parallelism do not tackle the long sequence length challenge as the sequence length dimension is unchanged.

The authors propose a systems level contribution that builds upon ring attention, which is a peer to peer communication method that allows for nearly infinite context lengths but has a high communication overhead. The authors propose a more sophisticated peer-to-peer communication scheme by grouping the GPUs into teams and dividing the peer-to-peer communication within these teams.

The authors include both theoretical analysis (Section 3.2.2) as well as extensive experimental results showing their approach outperforms Ring Attention under both strong and weak scaling for a GPT style transformer and a diffusion transformer.

Moreover, the authors have thoroughly sought to address reviewer concerns including both additional theory ( e.g. distinguishing between inter-node bandwidth and intra-node bandwidth) as well as including additional experiments (e.g. a comparison with Double Ring Attention).

Thus I recommend acceptance.